# Plasma Small Extracellular Vesicle Cathepsin D Dysregulation in *GRN/C9orf72* and Sporadic Frontotemporal Lobar Degeneration

**DOI:** 10.3390/ijms231810693

**Published:** 2022-09-14

**Authors:** Sonia Bellini, Claudia Saraceno, Luisa Benussi, Andrea Geviti, Antonio Longobardi, Roland Nicsanu, Sara Cimini, Martina Ricci, Laura Canafoglia, Cinzia Coppola, Gianfranco Puoti, Giuliano Binetti, Giacomina Rossi, Roberta Ghidoni

**Affiliations:** 1Molecular Markers Laboratory, IRCCS Istituto Centro San Giovanni di Dio Fatebenefratelli, 25125 Brescia, Italy; 2Service of Statistics, IRCCS Istituto Centro San Giovanni di Dio Fatebenefratelli, 25125 Brescia, Italy; 3Unit of Neurology V—Neuropathology, Fondazione IRCCS Istituto Neurologico Carlo Besta, 20133 Milan, Italy; 4Integrated Diagnostics for Epilepsy, Fondazione IRCCS Istituto Neurologico Carlo Besta, 20133 Milan, Italy; 5Department of Advanced Medical and Surgical Sciences, University of Campania “L. Vanvitelli”, 80131 Naples, Italy; 6MAC-Memory Clinic and Molecular Markers Laboratory, IRCCS Istituto Centro San Giovanni di Dio Fatebenefratelli, 25125 Brescia, Italy

**Keywords:** cathepsin D, frontotemporal lobar degeneration, endo-lysosomal pathway, GRN, C9orf72, extracellular vesicles, plasma, frontotemporal dementia, lysosomal protease

## Abstract

Emerging data suggest the roles of endo-lysosomal dysfunctions in frontotemporal lobar degeneration (FTLD) and in other dementias. Cathepsin D is one of the major lysosomal proteases, mediating the degradation of unfolded protein aggregates. In this retrospective study, we investigated cathepsin D levels in human plasma and in the plasma small extracellular vesicles (sEVs) of 161 subjects (40 sporadic FTLD, 33 intermediate/pathological *C9orf72* expansion carriers, 45 heterozygous/homozygous *GRN* mutation carriers, and 43 controls). Cathepsin D was quantified by ELISA, and nanoparticle tracking analysis data (sEV concentration for the cathepsin D level normalization) were extracted from our previously published dataset or were newly generated. First, we revealed a positive correlation of the cathepsin D levels with the age of the patients and controls. Even if no significant differences were found in the cathepsin D plasma levels, we observed a progressive reduction in plasma cathepsin D moving from the intermediate to *C9orf72* pathological expansion carriers. Observing the sEVs nano-compartment, we observed increased cathepsin D sEV cargo (ng/sEV) levels in genetic/sporadic FTLD. The diagnostic performance of this biomarker was fairly high (AUC = 0.85). Moreover, sEV and plasma cathepsin D levels were positively correlated with age at onset. In conclusion, our study further emphasizes the common occurrence of endo-lysosomal dysregulation in *GRN/C9orf72* and sporadic FTLD.

## 1. Introduction

Frontotemporal lobar degeneration (FTLD) is the second most common form of dementia after Alzheimer’s disease (AD) affecting people younger than 65 years old [1,2]. The major brain areas affected by FTLD are the frontal and temporal lobes, resulting in progressive behavioral disturbances and executive and language dysfunctions [3]. The main FTLD pathological hallmarks include protein aggregates, glia hyperproliferation and inflammation, lysosomal dysfunction, and neuronal loss. In particular, TAR DNA-binding protein (TDP)-43 is among the most common neuronal protein aggregates in FTLD, followed by microtubule-associated protein tau (MAPT) and fused in sarcoma (FUS) protein [4,5]. This genetic etiology has been revealed in 30–50% of FTLD patients with a positive family history of dementia [6,7,8]. The most common disease-causing genes are *MAPT* [9,10], progranulin (*GRN*) [11,12], and chromosome 9 open reading frame (*C9orf72*) [13,14]. The majority of *GRN* mutations are loss-of-function mutations, which introduce premature termination codons with the consequent degradation of the mutant RNA due to nonsense-mediated decay, resulting in a severe reduction in circulating progranulin, which is easily detectable by the plasma/serum dosage [15]. Additionally, missense mutations also lead to reduced functional progranulin by impairing its secretion or cleavage and/or increasing the level of misfolding [16,17]. Homozygous *GRN* mutations cause a complete loss of progranulin, leading to neuronal ceroid lipofuscinosis (NCL), a lysosomal storage disorder which shares some neuropathological features with *GRN*-associated FTLD [18]. The pathological expansions of a hexanucleotide repeat (>30 hexanucleotide G_4_C_2_ repeats) in the first intron/promoter of the *C9orf72* gene are among the most common causes of familial FTLD. Instead, intermediate expansions (12–30 hexanucleotide repeats) have a risk effect on familial/sporadic FTLD, influencing *C9orf72* expression and the disease phenotype according to the age at onset and clinical subtype [13,14,19].

Cathepsin D is a soluble lysosomal aspartic protease encoded by the *CTSD* gene. In the Golgi complex, it exists as a diglycosylated precursor form (pro-cathepsin D, 52 kDa), and the removal of its N-terminal propeptide results in a 48 kDa single chain intermediate active form. The final maturation step requires further processing in the lysosomes, generating the mature active lysosomal protease, composed of heavy (34 kDa) and light (14 kDa) chains linked by non-covalent interactions [20,21]. Cathepsin D is one of the most abundant lysosomal proteases, with a crucial role in autophagy and the endo-lysosomal system, mediating the degradation of unfolded protein aggregates. Intra-lysosomal proteolysis is essential for neuronal cell survival, and its persistent impairment favors the progressive accumulation of undigested autophagic substrates, leading to neurodegeneration [22]. In particular, several proteins (i.e., amyloid precursor, tau, α-synuclein, and lipofuscin), whose altered processing is associated with neurodegenerative diseases, are physiologic cathepsin D substrates and accumulate if they are not efficiently degraded [23,24,25,26]. Accordingly, several mutations and polymorphisms of *CTSD* have been linked with the pathogenesis of, or predisposition to, neurodegenerative diseases, such as NCL, AD, and Parkinson’s disease (PD) [27,28,29,30]. Impaired lysosomal proteolysis and decreased activity of the lysosomal enzyme cathepsin D were reported in fibroblasts lysates in heterozygous *GRN* mutation carriers [31] and in iPSC-derived human cortical neurons in frontotemporal dementia (FTD) patients harboring *GRN* mutations [32]. Furthermore, progranulin and, especially, its cleavage product granulin E were demonstrated to be able to increase cathepsin D activity [33,34]. Thus, cathepsin D activity reduction due to the loss of progranulin was suggested to contribute to both the FTD and NCL pathologies in a dose-dependent manner [32]. In agreement with findings on fibroblasts and neurons in *GRN* mutation carriers, cathepsin D activity was also significantly reduced in *c9orf72*^−/−^ mice macrophages [35], in the frontal cortex of *Grn*^−/−^ mice [36], and in *Grn* KO primary mouse microglia compared to the wild type [37]. In contrast, cathepsin D levels were significantly elevated in *Grn*
^−/−^ mouse brains [38,39,40], in the microglia of *Grn* KO mice [41], and also in frontal cortex of both FTLD-TDP patients with the *GRN* mutation and NCL patients [39], suggesting a possible compensatory mechanism in response to the loss of progranulin. Finally, the cathepsin D content was significantly increased in the neural-derived plasma exosomes of behavioral FTD patients compared to the controls as a possible result of autophagocytic-lysosomal dysfunction in FTLD [42]. Endo-lysosomal dysregulation has already been suggested to be a pathological feature in *GRN/C9orf72*-associated FTLD [43,44]. Extracellular vesicles (EVs) are membranous particles that are naturally released from cells, comprising exosomes (endosomal origin) and microvesicles (plasma-membrane-derived) [45]. EVs have been demonstrated to play a crucial role in intercellular communication and molecular transfer, but also in reducing the burden of the lysosomal storage material or toxic proteins, thus sustaining their either protective or pathogenic role [46,47]. In a recent study of ours, we reported a decrease in the small EV (sEV) concentration and an increase in the sEV size in human plasma samples from *GRN/C9orf72*-associated and sporadic FTLD patients, thus suggesting that a common molecular mechanism is altered in both genetic and sporadic FTLD [48].

In this study, we expanded on our studies of the endo-lysosomal pathway in both *GRN/C9orf72*-genetic and sporadic FTLD, investigating, in particular, the lysosomal protease cathepsin D in the sEVs and plasma.

## 2. Results

### 2.1. sEV Cathepsin D Dysregulation in Genetic and Sporadic FTLD

In this retrospective study, we investigated cathepsin D levels in 161 subjects (118 patients and 43 controls). In detail, patients with sporadic FTLD (n = 40), with FTLD due to C9orf72 expansion (n = 9, C9orf72 intermediate expansion carriers, C9orf72 Int.; n = 24, C9orf72 pathological expansion carriers, C9orf72 Pat.), with FTLD due to pathogenic mutations in GRN (n = 42, GRN heterozygous mutation carriers, GRN+ Het.), with NCL due to pathogenic mutations in GRN (n = 3, GRN homozygous null mutation carriers, GRN+ Homo.), and subjects with normal cognitive function, as a control group (n = 43, Ctrl), were included. Demographic and clinical characteristics are reported in Table 1.

The isolated sEVs were CD9+ (a tetraspanin), TSG101+ and Flotillin-1 + (cytosolic proteins recovered in EVs), and Calnexin negative (endoplasmic reticulum residential protein, absent in EVs) (Appendix A). The cathepsin D precursor and intermediate forms, but not mature cathepsin D, were detected in human plasma sEVs (Figure 1). The sEV concentration data were extracted from our previously published larger dataset [48] or were newly generated. Plasma sEV concentrations (sEVs/mL) of the samples included in the present study are reported in Table 1.

The cathepsin D quantification in the C9orf72 Int., C9orf72 Pat., GRN+ Het., GRN+ Homo., Sporadic FTLD, and Ctrl groups was performed by an ELISA assay (Table 1). First, we evaluated the association between the demographic variables (age, sex) and sEV cathepsin D. Age was positively correlated with sEVs cathepsin D levels normalized according to the sEV concentration (cathepsin D concentration per sEV, ng/sEV), both in the whole patient group (PTS, including C9orf72 Int., C9orf72 Pat., GRN+ Het., GRN+ Homo., and Sporadic FTLD; Spearman r = 0.261, *p* = 0.004) and in the Ctrl (Spearman r = 0.474, *p* = 0.001) (Figure 2). The same is for the total cathepsin D content in the sEVs (ng/mL, not normalized values) (Appendix A). No sex-related differences were observed in terms of the sEV cathepsin D levels among the whole group (Appendix A) or the Ctrl and PTS groups, individually (Appendix A).

The cathepsin D concentration per sEV (ng/sEV) was significantly increased in all patient groups (C9orf72 Int., C9orf72 Pat., GRN+ Het., GRN+ Homo., and Sporadic FTLD) compared with the Ctrl (Table 1 and Figure 3a, global *p* < 0.001, generalized linear model, adjusted for age and sex). Considering the total cathepsin D content in the sEVs (ng/mL, not normalized values), a statistically significant difference was found between the investigated groups (Table 1 and Figure 3b, global *p* = 0.019, generalized linear model, adjusted for age and sex), but there were no statistically significant differences between the pair of groups after the Bonferroni correction for multiple comparisons.

Moreover, we found that the sEV cathepsin D concentration was positively correlated with age at onset among the whole patient group (Spearman r = 0.262, *p* = 0.011) (Figure 4).

To estimate the diagnostic performance of the cathepsin D concentration per sEV, ROC analyses were performed (Figure 5). The reliability of the cathepsin D concentration per sEV in discriminating between the PTS and Ctrl groups was fairly high (AUC = 0.85), with a sensitivity of 75.4% and specificity of 76.7%, considering the cut-off point of 1.72 × 10^−11^ ng/sEV. ROC curves for the Ctrl vs. Genetic FTLD (C9orf72 Int., C9orf72 Pat., GRN+ Het., and GRN+ Homo.) and Ctrl vs. Sporadic FTLD had similar AUCs (0.83 and 0.89 respectively), using the DeLong test for the AUC comparison (*p* value = 0.210).

### 2.2. Plasma Cathepsin D in Sporadic and Genetic FTLD

Furthermore, cathepsin D levels were analyzed using the plasma of the PTS and Ctrl groups. As observed for sEV cathepsin D, age was positively correlated with plasma cathepsin D in both the PTS and Ctrl groups (PTS Spearman, r = 0.254, *p* = 0.006; Ctrl r = 0.471, *p* = 0.002; Appendix A). No sex-related differences were observed in the Ctrl and PTS (Appendix A). As shown in Table 1 and in Appendix A, no statistically significant differences were found among the investigated groups (global *p* = 0.058, generalized linear model, adjusted for age and sex). Accordingly, the diagnostic performance of plasma cathepsin D in discriminating between the PTS and the Ctrl group was poor (AUC = 0.63), with a sensitivity of 60.2% and specificity of 64.3%, considering the cut-off point of 151.8 ng/mL (Appendix A).

When we investigated the putative dose effects of the *C9orf72* expansions (Figure 6a) and *GRN* null alleles (Figure 6b) on the cathepsin D plasma levels, we revealed a progressive reduction in the plasma cathepsin D concentration moving from C9orf72 Int. to C9orf72 Pat. (Ctrl vs. C9orf72 Int: −11.18%; Ctrl vs. C9orf72 Pat: −22.36%; *p* = 0.008, generalized additive linear model, adjusted for age and sex) (Figure 6a). No significant differences were reported in terms of the *GRN* null alleles (Figure 6b).

Moreover, considering the whole patient group, we found that the levels of plasma cathepsin D were positively correlated with age at onset (Spearman r = 0.291, *p* = 0.004) (Figure 7).

## 3. Discussion

Emerging data suggest the roles of endo-lysosomal dysfunctions in FTLD and in neurodegenerative diseases in general [43,44,46]. Cathepsin D is one of the major lysosomal proteases and is responsible for the degradation of proteins and organelles through the autophagy–lysosomal system [49].

In this study, we provide evidence of an alteration in the lysosomal protease cathepsin D in both sporadic and genetic FTLD. Recent studies investigated the role of cathepsin D as a plasma biomarker of PD and AD [50,51,52]. In PD, a reduction in plasma cathepsin D levels was suggested to be a biomarker in the early stages of the disease and was not correlated with age at onset, disease duration, or specific endophenotypes [51]. In AD, the plasma cathepsin D levels were demonstrated to be reduced and negatively correlated with disease severity [50]. However, a previous study in AD described an increase in cathepsin D at different disease stages [52]. Thus, more studies are needed in order to clarify the role of cathepsin D as a blood biomarker of dementias.

In the present study, we assessed cathepsin D levels in the plasma and plasma sEVs of a relatively large group of FTLD patients, including sporadic patients, genetic patients (intermediate/pathological C9orf72 expansion carriers, GRN heterozygous mutation carriers), and GRN homozygous mutation carriers affected by NCL, a lysosomal storage disorder. First, we revealed a positive correlation of cathepsin D levels with age among both the patients and controls. This is in line with early studies on rats, in which cathepsin D levels were increased in aged brains [53,54]. Cytosolic cathepsin D was selectively increased not only in the rat cerebral cortex but also in the hippocampus, cerebellum, kidneys, liver, and spleen [55], suggesting that it is a marker of aging, playing a potential protective role. In our study, no alterations were detected in the circulating cathepsin D plasma levels in the patients compared with the controls. Even if no significant differences were found among the investigated groups, we observed a progressive reduction in plasma cathepsin D in the C9orf72 cases, moving from C9orf72 Int. to C9orf72 Pat., indicating a putative dose effect of the C9orf72 expansions on the cathepsin D plasma levels.

Examining the extracellular vesicles in the blood, we observed an increased cathepsin D sEV cargo in the sporadic and genetic FTLD groups. A significant increase in the cathepsin D concentration per sEV (normalized values according to the sEV concentration) was found in all patient groups compared to the controls. The diagnostic performance of the cathepsin D concentration per sEV as a parameter in discriminating between the patients and healthy controls was fairly high (AUC = 0.85), with a sensitivity of 75.4% and specificity of 76.7%, considering the cut-off point of 1.72 × 10^−11^ ng/sEV. In contrast, in our samples, the diagnostic performance of plasma cathepsin D in discriminating between the PTS and Ctrl groups was poor (AUC = 0.63), thus supporting the need for the isolation of EVs as a source of biomarkers. We have previously demonstrated that genetic and sporadic FTLD share lower sEV concentrations and higher sEV sizes. The diagnostic performance of the sEV concentration/size ratio was high (AUC = 0.91), with a sensitivity 85.3% and a specificity 81.4% [48]. Thus, although the cathepsin D concentration per sEV showed a fair diagnostic performance, the plasma sEV concentration/size ratio may represent a better biomarker. However, we believe that the evidence reported herein has added value for the comprehension of the mechanism underlying the pathology. The observed increase in the sEV cathepsin D levels is in agreement with a previous study on the neural-derived plasma exosomes (a subtype of EVs of endosomal origin) of sporadic behavioral FTD patients, where the cathepsin D content was significantly increased compared to that of the controls [42]. Similarly, an increase in cathepsin D was observed in the frontal cortexes of both FTLD patients with a *GRN* mutation and NCL patients [39], as well as in *Grn*^−/−^ mouse brains [38,39,40], suggesting a possible common compensatory response to endo-lysosomal dysfunction. Moreover, in our study, the sEV cathepsin D level (as well as plasma cathepsin D level) was positively correlated with the age at onset, indicating that lower cathepsin D levels could lead to an earlier disease onset. By comparison, cathepsin D in the circulating sEVs was not associated with sex, even though it is known that it is a type of lysosomal protease induced by estrogens [56]. Moreover, investigating the cathepsin D forms in sEVs by Western blot analyses, we noticed that, both among the controls and patients, the cathepsin D precursor was the major form of cathepsin D present in the human plasma sEVs, while the mature form was not detected.

Breakthrough molecular studies have suggested the roles of endo-lysosomal dysfunctions in *GRN/C9orf72* FTLD [43,44], with the accumulation of lipofuscin in the brain and an impaired activity of the lysosomal enzymes [31,39]. Moreover, we recently reported that genes involved in the endo-lysosomal pathway (i.e., protein sorting/transport, clathrin-coated vesicle uncoating, and lysosomal enzymatic activity regulation) might be involved in neurodegenerative diseases, including FTLD, even if the ways in which these alterations impact the EV release/composition, brain–periphery communication, and neurodegeneration are still a matter of discussion [57].

To the best of our knowledge, this is the first study reporting an alteration in the levels of plasma sEV-associated cathepsin D in *GRN/C9orf72*-genetic FTLD patients and describing an alteration in cathepsin D trafficking that is common to both the genetic and sporadic forms.

The most well-established and clinically approved approach to treat lysosomal storage disorders is enzyme replacement therapy (ERT), aiming to replace the defective hydrolase with an exogenously applied recombinant protein [58]. The therapeutic potential of ERT with recombinant human pro-cathepsin D has recently been demonstrated in a murine pre-clinical model of NCL [59]. Based on the role played by cathepsin D in age-associated neurodegenerative diseases, as in NCL, cathepsin D may be a promising therapeutic target [60].

A limitation of this study is the small number of patients included in the C9orf72 Int. and GRN+ Homo. groups, which could have masked further significant differences between the investigated groups. This is a pilot study, and further validation using a larger and/or independent cohort is warranted.

In conclusion, our study further emphasizes the common role of endo-lysosomal dysregulation in *GRN/C9orf72* and sporadic FTLD, further highlighting the relevance of the endo-lysosomal pathway in neurodegenerative dementias.

## 4. Materials and Methods

### 4.1. Subjects

The project included a total of 161 subjects (118 patients and 43 controls). In detail, patients with sporadic FTLD (n = 40), with FTLD due to *C9orf72* expansion (n = 9 C9orf72 Int. and n = 24 C9orf72 Pat.), with FTLD due to pathogenic mutations in *GRN* (n = 42 GRN+ Het.), with NCL due to pathogenic mutation in *GRN* (n = 3 GRN+ Homo.), and subjects with normal cognitive function, as a control group (n = 43, Ctrl), were included in the present retrospective study. Patients were enrolled at the MAC Memory Clinic IRCCS Fatebenefratelli, Brescia, and at the Neurology 5/Neuropathology Unit, IRCCS Besta, Milan. The clinical diagnosis of FTLD was performed according to international guidelines [61,62]. The diagnosis of NCL was performed by the ultrastructural examination of a skin biopsy [18]. Participants signed an informed consent form for the blood collection and storage in a biobank/biorepository, as approved by the local ethics committee (approval numbers 2/1992; 26/2014). *C9orf72* and *GRN* genetic screening was performed previously, as described [19,63,64]. The study protocol was approved by the local ethics committee, “Comitato Etico IRCCS San Giovanni di Dio Fatebenefratelli”, of the IRCCS Centro San Giovanni di Dio Fatebenefratelli, Brescia (prot. No. 44/2018).

### 4.2. Small EV Isolation and Characterization

Total plasma sEV isolation was performed using plasma with the Total Exosome Isolation Kit (Invitrogen^TM^, Waltham, MA, USA), following the manufacturer’s protocol. Briefly, 125 µL of plasma was centrifuged at room temperature (RT) at 2000× *g* for 20 min and at 10,000× *g* for 20 min to remove cells and debris. The supernatants were then incubated for 10 min at RT with 62.5 µL of 0.2 µM-filtered 1X phosphate-buffered saline (PBS) and 37.5 µL of exosome precipitation reagent. After 10 min of incubation at RT, the suspensions were centrifuged at 10,000× *g* for 5 min. The sEV pellets were lysed with 60 µL of ice-cold exosome resuspension buffer (Total Exosome RNA and Protein Isolation Kit, InvitrogenTM, Waltham, MA, USA), and stored at −20 °C for the Western blotting and biochemical analyses. The nanoparticle tracking analysis (NTA) data (sEV concentration and size) were extracted from our previously published larger dataset, which is openly available in the Mendeley Data Repository (at doi:10.17632/kds9sb4z6t.2), or generated as previously described [48]. The plasma sEV concentrations (sEVs/mL) of the samples included in the present study were as follows: Ctrl 2.3 × 10^11^ ± 1.1 × 10^11^, C9orf72 Int. 1.5 × 10^11^ ± 7.1 × 10^10^, C9orf72 Pat. 1.1 × 10^11^ ± 5.3 × 10^10^, GRN+ Het. 9.7 × 10^10^ ± 4.3 × 10^10^, GRN+ Homo. 8.3 × 10^10^ ± 1.6 × 10^10^, and Sporadic FTLD 1.1 × 10^11^ ± 6.3 × 10^10^. Plasma sEV size (nm) values were as follows: Ctrl 115.1 ± 14.5, C9orf72 Int. 137.4 ± 31.0, C9orf72 Pat. 136.9 ± 10.0, GRN+ Het. 144.6 ± 16.2, GRN+ Homo. 136.3 ± 14.3, and Sporadic FTLD 135.1 ± 11.6.

For the sEV characterization, the Flotillin-1, TSG101, CD9, and Calnexin expression were analyzed in the sEVs by Western blotting, according to standard protocols. Briefly, the resuspended sEVs were separated using Bolt^TM^ 4–12% Bis-Tris Plus Gels (InvitrogenTM, Waltham, MA, USA) with MOPS SDS running buffer (Invitrogen^TM^, Waltham, MA, USA) and electro-transferred onto nitrocellulose membranes (ThermoFisher Scientific, Waltham, MA, USA) for 90 min at 90 V. Incubation with primary antibodies was performed overnight at +4 °C (anti-Flotillin-1, 1:500; anti-TSG101, 1:500; anti-CD9, 1:500; Abcam, Cambridge, UK; anti-Calnexin, 1:1000; BD Biosciences, Franklin Lakes, NJ, USA), followed by incubation with horseradish-peroxidase-conjugated secondary antibodies, 1:10,000 (Invitrogen^TM^, Waltham, MA, USA), for 1 h at +37 °C. Immuno-positive bands were detected by ultra-sensitive enhanced chemiluminescence (ThermoFisher Scientific, Waltham, MA, USA) according to the manufacturer’s instructions. The isolated sEVs were CD9+ (a tetraspanin), TSG101+ and Flotillin-1 + (cytosolic proteins recovered in EVs), and Calnexin negative (endoplasmic reticulum residential protein, absent in EVs) with a size of <200 nm (small EVs, [45]) (Appendix A). The sEV Cathepsin D forms were characterized by Western blotting, following standard protocols and using the rabbit monoclonal anti-Cathepsin D (Abcam, Cambridge, UK) as the primary antibody.

### 4.3. Biochemical Analyses

The cathepsin D quantification of the human non-EV-depleted plasma and sEVs was performed using the Human Cathepsin D ELISA kit (Sigma-Aldrich, St. Louis, MO, USA), following the manufacturer’s instructions. Plasma sEV samples were diluted at a ratio of 1:20 while plasma samples were diluted at 1:100. All analyses were performed in duplicate. The sEV-associated cathepsin D concentration data were normalized (or not) according to the sEV concentration (cathepsin D concentration per sEV, ng/EV, and total cathepsin D concentration in sEVs, ng/mL, respectively).

### 4.4. Statistical Analysis

The normality assumption of the continuous variables was assessed by the Kolmogorov–Smirnov test. One-way ANOVA, with Bonferroni post hoc tests, was used for the comparison of the normally distributed demographic and biological variables between the groups. The Mann–Whitney test was used for the comparison of non-normally distributed variables between the two groups. The chi-square test (or Fisher’s exact test, when appropriate) was used to assess the association between the categorical variables and the group variables. Generalized linear models or generalized additive linear models (for non-normally distributed variables) with a gamma or inverse gaussian distribution were used for the comparison of the target variables between the groups (adjusting for age and sex). Moreover, Bonferroni’s correction was applied to the *p* values of pairwise comparisons. Spearman correlations were performed for the cathepsin D quantification and age/age at onset. The diagnostic performances of the cathepsin D concentration per sEV and plasma cathepsin D variables were assessed by areas under the curve (AUC), obtained by the receiver operating characteristic (ROC). The comparison of the AUC results was assessed by the DeLong test. All analyses were performed using SPSS software (V27) or R software, and the significance was set as 0.05.

## Figures and Tables

**Figure 1 ijms-23-10693-f001:**
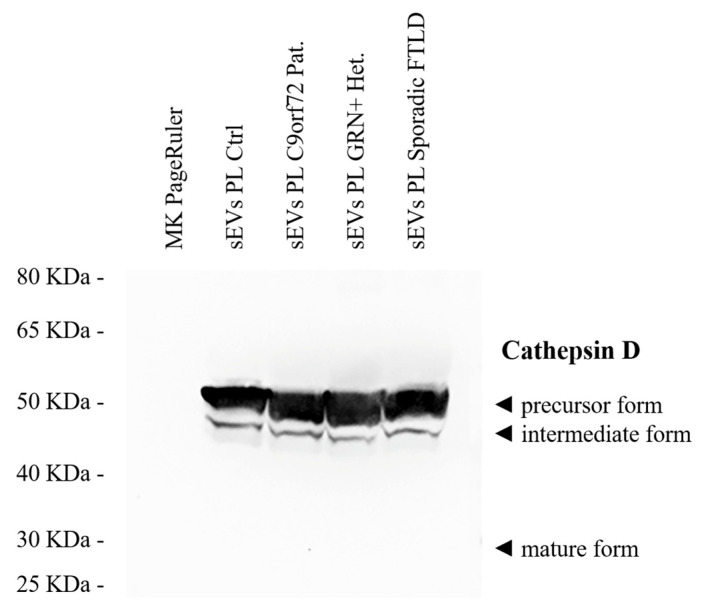
Cathepsin D forms in human plasma sEVs. Western blot analysis of the representative plasma sEV samples from Ctrl., C9orf72 Pat., GRN+ Het., and sporadic FTLD patients.

**Figure 2 ijms-23-10693-f002:**
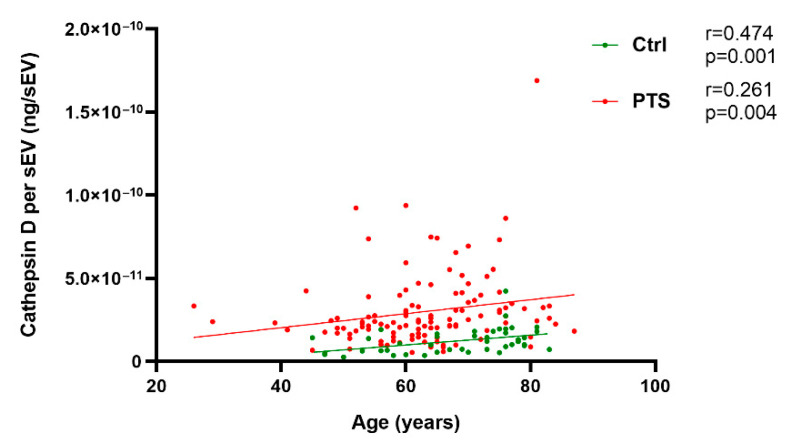
Correlation of cathepsin D concentration per sEV (ng/sEV) with age among the controls (Ctrl, in green) and patients (PTS, in red). Spearman correlations r and *p* values are shown in the panel.

**Figure 3 ijms-23-10693-f003:**
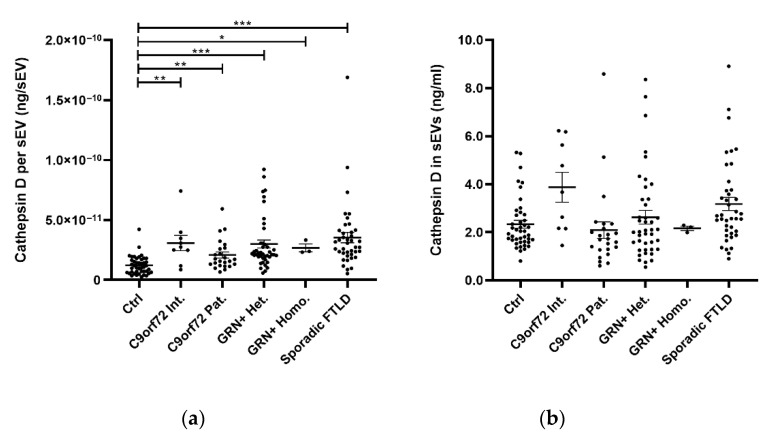
Cathepsin D (**a**) concentration per sEV (ng/sEV); (**b**) total concentration in sEVs (ng/mL), non-normalized values in the Ctrl, C9orf72 Int., C9orf72 Pat., GRN+ Het., GRN+ Homo., and Sporadic FTLD. Mean ± SEM; * *p* < 0.05, ** *p* < 0.01, *** *p* < 0.001. Dot plots represent raw data, while the post hoc *p* values were obtained using a generalized linear model with a gamma distribution (**a**) or inverse gaussian distribution (**b**), both adjusted for age and sex.

**Figure 4 ijms-23-10693-f004:**
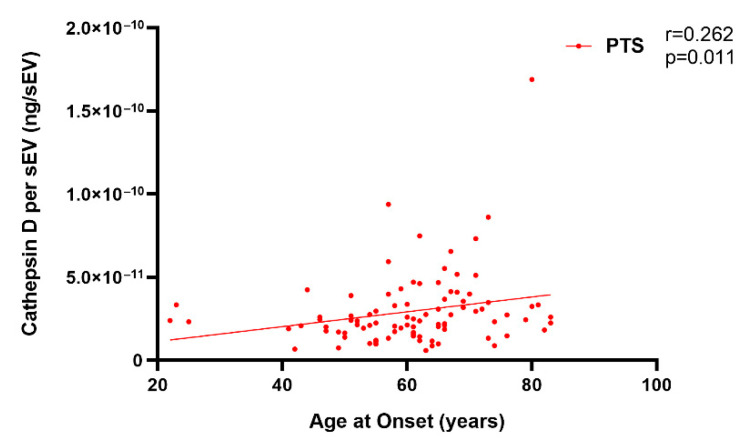
Correlation of the cathepsin D concentration per sEV (ng/sEV) with age at onset in the PTS group. The Spearman correlation r and *p* value are shown in the panel.

**Figure 5 ijms-23-10693-f005:**
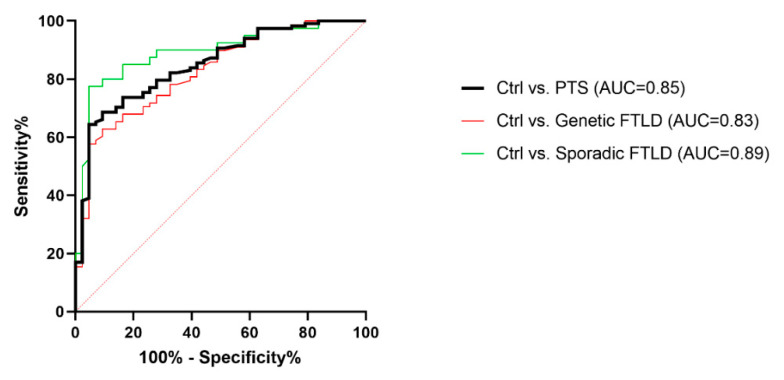
ROC curves for the cathepsin D concentration per sEV (ng/sEV). Cathepsin D concentration per sEV was used to evaluate the discrimination of the PTS from the Ctrl group. AUC Ctrl vs. PTS 0.85 (black); AUC Ctrl vs. Genetic FTLD 0.83 (red); AUC Ctrl vs. Sporadic FTLD 0.89 (green). AUC comparison with the DeLong test, *p* = 0.210.

**Figure 6 ijms-23-10693-f006:**
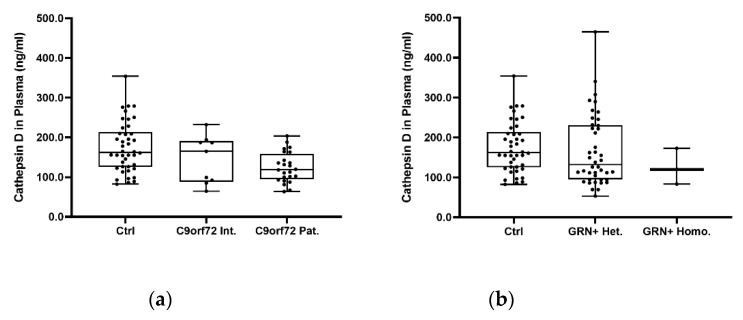
Cathepsin D concentration in the plasma (ng/mL) of the (**a**) Ctrl, C9orf72 Int., and C9orf72 Pat., and in (**b**) Ctrl, GRN+ Het., and GRN+ Homo. Box plots represent raw data, while the *p* values were obtained using a generalized additive linear model with an inverse gaussian distribution adjusted for age and sex.

**Figure 7 ijms-23-10693-f007:**
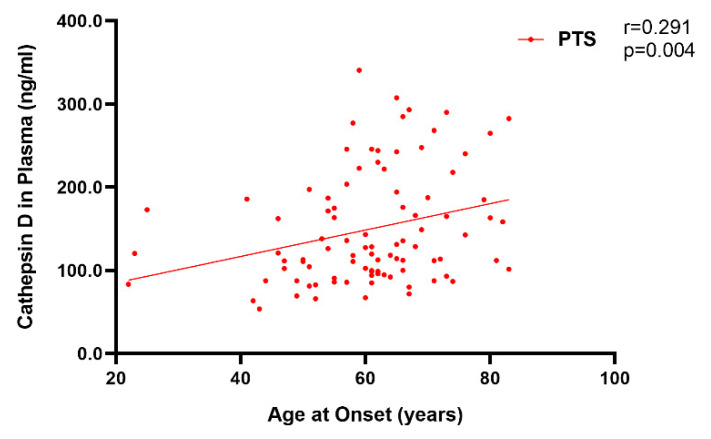
Correlation of plasma cathepsin D (ng/mL) with age at onset in the PTS group. Spearman correlation r and *p* value are shown in the panel.

**Table 1 ijms-23-10693-t001:** Clinical, demographic, and biological characteristics of the patients and controls included in the study.

	Ctrl	C9orf72 Int.	C9orf72 Pat.	GRN+ Het.	GRN+ Homo	Sporadic FTLD	*p* Value
N.	43	9	24	42	3	40	
Sex (% female)	72.1	44.4	45.8	42.9	33.3	65.0	0.053 ^a^
Age, years	68.1 ± 10.6	67.2 ± 5.2	62.7 ± 10.4	60.1 ± 7.6	31.3 ± 6.8	67.7 ± 10.4	<0.001 ^b^
Age at onset, years	/	64.1 ± 6.2	60.6 ± 10.6	58.9 ± 8.1	23.3 ± 1.5	64.4 ± 10.8	<0.001 ^b^
EV conc., EVs/ml	2.3 × 10^11^ ± 1.1 × 10^11^	1.5 × 10^11^ ± 7.1 × 10^10^	1.1 × 10^11^ ± 5.3 × 10^10^	9.7 × 10^10^ ± 4.3 × 10^10^	8.3 × 10^10^ ± 1.6 × 10^10^	1.1 × 10^11^ ± 6.3 × 10^10^	<0.001 ^c^
Cathepsin D per sEV, ng/sEV ^#^	1.2 × 10^−11^ ± 7.6 × 10^−12^	3.1 × 10^−11^ ± 1.9 × 10^−11^	2.1 × 10^−11^ ± 1.3 × 10^−11^	3.0 × 10^−11^ ± 2.2 × 10^−11^	2.7 × 10^−11^ ± 5.6 × 10^−12^	3.5 × 10^−11^ ± 2.8 × 10^−11^	<0.001 ^c^
Cathepsin D in sEVs, ng/mL ^#^	2.3 ± 1.1	3.9 ± 1.9	2.1 ± 1.7	2.6 ± 1.8	2.2 ± 0.2	3.2 ± 1.8	0.019 ^d^
Cathepsin D plasma, ng/mL ^#^	175.5 ± 63.5	145.1 ± 60.2	124.5 ± 38.0	166.5 ± 89.3	125.5 ± 45.0	157.1 ± 69.2	0.058 ^d^

Ctrl = controls; C9orf72 Int. = C9orf72 intermediate expansion carriers; C9orf72 Pat. = C9orf72 pathological expansion carriers; GRN+ Het. = GRN heterozygous null and missense mutation carriers; GRN+ Homo. = GRN homozygous null mutation carriers; Sporadic FTLD = patients with sporadic frontotemporal lobar degeneration. ^a^ Chi-square test; ^b^ one-way ANOVA test; ^c^ generalized linear model with gamma distribution (adjusted for age and sex); ^d^ generalized linear model with inverse gaussian distribution (adjusted for age and sex). ^#^ Age is significant for this model. Mean ± standard deviation.

## Data Availability

The data presented in this study are openly available in the Zenodo Data Repository at doi:10.5281/zenodo.6958338 [65] (dataset creation date: 3 August 2022).

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
