# Peer review of "Plasma Small Extracellular Vesicle Cathepsin D Dysregulation in GRN/C9orf72 and Sporadic Frontotemporal Lobar Degeneration"

_ijms, 2022, doi:10.3390/ijms231810693_

Round 1
Reviewer 1 Report
Review of a manuscript:” Plasma Small Extracellular Vesicles Cathepsin D Dysregulationin GRN/C9orf72 and Sporadic Frontotemporal Lobar Degeneration” by Sonia Bellini and coauthors submitted to IJMS.
Frontotemporal lobar degeneration is a prevalent form of dementia which is often found in patients younger than 65 years. One of the pathological features of frontotemporal lobar degeneration is endo-lysosomal dysregulation in which cathepsin D plays an important role. The authors investigated cathepsin D levels in human plasma and in extracellular vesicles of frontotemporal lobar degeneration patients and control individual in an attempt to clarify the role of this enzyme in pathology. This is important topic and the results of the investigation will be interesting for the readers of IJMS.
The following corrections and additions should be made:
Introduction:
Line 55: ”Additionally, missense mutations also lead to reduced functional progranulin by impairing secretion, misfolding or cleavage [16,17].” The sense of this sentence is in clear. Does it mean that missense mutations impair misfolding causing correct folding? Or vice versa they increase the level of misfolding? This should be clarified.
Lines 96-97: ”Extracellular vesicles (EVs) are membranous particles naturally released from cells, comprising exosomes (endosomal origin) and microvesicles (plasma membrane derived) [45].” The authors should add here the following reference: ”gamma-Synuclein: Seeding of α-Synuclein Aggregation and Transmission Between Cells. Biochemistry, 2012; 51(23):4743-54”.
Results
Lines 111-118. In this very long sentence the authors should delete “in the present retrospective study” which they used in a previous sentence.
Lines 121-122: ”Cathepsin D precursor and intermediate forms were detected in human plasma sEVs. The mature cathepsin D form was not detected in sEVs (Figure 1).” These two sentences may be combined in one as follows:”Cathepsin D precursor and intermediate forms, but not mature cathepsin D were detected in human plasma sEVs (Figure 1).”
Materials and Methods
Line 286. “The study protocol was approved by the local ethics committee (Prot. N. 44/2018)”. The authors should replace “local” on the exact name of the Committee.
Lines 314-315:”Incubation with primary antibodies was performed overnight at + 4 °C (anti-Flotillin-1, anti-TSG101, anti-CD9, Abcam, Cambridge, UK; anti-Calnexin, BD Biosciences, Franklin Lakes, NJ, USA), followed by incubation with horseradish peroxidase conjugated secondary antibodies (InvitrogenTM, Waltham, MA, USA) for 1 h at +37°C.”
The dilutions of primary and secondary antibodies should be given.
Discussion
Lines 202-203: “Cathepsin D is one of the major lysosomal proteases, and is responsible for the degradation of proteins and organelles through the autophagy-lysosomal system.”
Lines 259-260: ”The most established and clinically approved approach to treat lysosomal storage disorders is the enzyme replacement therapy (ERT) aiming to replace the defective hydrolase with an exogenously applied recombinant protein”
The authors should add references for these two sentences.
Reviewer 2 Report
In the submitted manuscript, Bellini et al. present data showing that plasma extracellular vesicle associated Cathepsin D levels display high level diagnostic performance as a biomarker for both genetic and sporadic FTLD. The authors first show a strong positive correlation of that plasma extracellular vesicle associated Cathepsin D levels with age. this also highlights a major limitation of the study since the genetic FTLD patients a significantly younger that the healthy controls. The authors need to discuss if this difference in age was corrected for or how these results should be interpreted. They show a significance predictive power of this marker and they should discuss this performance compared to other potential biomarkers. The use of an independent cohort to validate their ROC analysis would improve the impact of the manuscript. Furthermore, was a ROC analysis performed on plasma Cathepsin D levels. Showing insignificant predictive power could further support the need for isolation of extracellular vesicles as a biomarker. The results for plasma cathepsin D levels needs minor clarification because it is unclear if this is from EV depleted plasma or this plasma also contains EVs. A small summary of the EV characterization would be a good addition to the beginning of the Results with a citation to the article where these results are displayed. In the discuss the authors should discuss in more detail the origin of these EVs. Do they hypothesis that these are brain derived or peripheral EVs? Overall, the article provides the data from a retrospective study for a potential biomarker of FTLD which could be impactful for the field if validated.
Reviewer 3 Report
This article entitled “Plasma Small Extracellular Vesicles Cathepsin D Dysregulation
in GRN/C9orf72 and Sporadic Frontotemporal Lobar Degeneration” by Bellini et al. investigated relation between the amount of CatD and age in both control subjects and patients carry various genetic risks in CatD/lysosomal function which are implicated in FTLD, the second leading cause of dementia after Alzheimer’s disease. Although the previous studies with AD patients were somewhat controversial, the current study clearly show robust positive correlation of CatD vs age and elevated CatD amounts in patients carrying lysosomal genetic risks. The conclusion of this study clearly has a great impact on the field of age-related neurodegeneration in general. Unfortunately, the control group is overwhelmingly dominated by female, which brings its stats almost significant (p=0.053). It is rather understandable that numbers of some groups, especially GRN homo were small due to rareness, which may have masked the effect seen in other groups, as the authors acknowledged. However, there is no good reason to have the control group with obviously unmatched sex ratio. Having the sex ratio matched control group is critical for the whole study, because it is what everything else is compared with.
Round 2
Reviewer 3 Report
The authors fully addressed the concern by comparing male and female subjects, demonstrating that there is no sex differences. The results were appropriately presented in the new supplemental data. The manuscript is ready for publication.